# Findings on Ad Hoc Contractions

**Sing Choi [†] and Kazem Taghva \*,[†]**

Department of Computer Science, University of Nevada, Las Vegas, NV 89154, USA; chois1@unlv.nevada.edu
\* Correspondence: kazem.taghva@unlv.edu
† These authors contributed equally to this work.

**Abstract:** Abbreviations are often overlooked, since their frequency and acceptance are almost second nature in everyday communication. Business names, handwritten notes, online messaging, professional domains, and different languages all have their own set of abbreviations. The abundance and frequent introduction of new abbreviations cause multiple areas of overlaps and ambiguity, which mean documents often lose their clarity. We reverse engineered the process of creating these ad hoc abbreviations and revealed some preliminary statistics on what makes them easier or harder to define. In addition, we generated candidate definitions for which it proved difficult for a word sense disambiguation model to select the correct definition.

**Keywords:** abbreviations; contractions; TF-IDF; ambiguous; definitions

## 1. Introduction

Abbreviations have widespread, everyday usage across multiple languages and domains. Their ease of use and time-saving benefits have incentivized their utilization in both professional and casual contexts. An analysis by Barnett and Doubleday was performed on the abstracts and titles of scientific literature from the years 1950 to 2019 on the usage of acronyms. Their results showed an increase in the use of acronyms from 0.7 per 100 words in 1950, to 2.4 per 100 words in 2019 [1].

The increasing amount of abbreviations in the professional sciences alone causes clarity issues and misunderstandings when trying to interpret the long form definitions. A survey done by Sheppard et al. [2] found over 2286 abbreviations being used in 25 clinical handout sheets. According to Tariq and Sharma, within the last twenty years, approximately 7000 to 10,000 people have died due to medical mistakes every year in the United States alone.Among these errors, misunderstanding about abbreviations have contributed to roughly 5% of these casualties. [3]. Minimizing the clarity issue with abbreviations could save a large portion of those documented 5% of lives, and researchers Liu et al. showed that expanding the definitions of abbreviations overwhelmingly increased the patient comprehension in regards to their health records [4].

While abbreviations can pose an immediate and fatal danger in the medical field, ambiguity regarding them can plague every text-related field, including computer science. Source code utilizes many creative shortenings of words and phrases to facilitate typing speed. Deployment of incorrect abbreviations or their over-usage can severely hurt the readability and understanding [5] of code. While researchers Jiang et al. found that overly lengthy words in code were detrimental to reader comprehension, and recommended the frequent usage of abbreviations to improve understanding, Hales et al. made the argument that this leads to more instances of misuse. This, when paired with the usage of esoteric jargon, can lead to readers becoming alienated and overwhelmed with the text, increasing confusion and misunderstandings [6].

First, we must lay a foundation of what acronyms and abbreviations are, before modifying their properties to satisfy their professional usage. Acronyms are usually a type of abbreviation where a phrase or sequence of words are shortened into the initials

of each major word (e.g., DOE stands for Department of Energy). An abbreviation is generally a relative term used to describe the shortening of a word using contractions or contracting parts of the word (e.g., dept stands for department). Each type of abbreviation will often follow a loose set of rules and cues that hint the reader that it is an abbreviation. Taghva and Gilbreth created the first rule-based machine algorithm to detect acronyms. This algorithm operated under rules of identification where the acronym length must be at least three letters long and its full nomenclature or definition must be provided within a certain window of text [7]. This later developed into a machine learning approach, utilizing HMMs (Hidden Markov Models), to accurately select the definition within a proportionate window size [8]. In a later section, we will modify some of these general purpose rules to better suit the medical domain and to suit contraction-type abbreviations.

Currently, there are three main approaches to handling abbreviations with regards to machine reading comprehension. The first approach involves rule-based/statistical models, such as HMMs or decision trees, which require both understanding the cues and inferences of the document, to select the best definition. These approaches often do well when data are structured and uniform, but fail in real-life scenarios, because abbreviations do not have a uniform set of rules that every author follows. In particular, these methods require some sort of regulation of their usage, such as employing all capital letters or including a definition in parentheses to indicate the existence of an acronym. Certain models use prior knowledge of context to determine the best expansion or try to find its definition nearby.

The second approach is through maximizing word sense disambiguation (WSD). This approach typically utilizes the context to determine the meaning from a set of potential definitions and selects the one which makes the most sense. Researchers such as Sultan et al. have considered parts-of-speech (POS) to determine whether or not definitions fit their grammatical place. In addition to their usage of POS, the authors also created pipelines of alignment modules to determine how closely related certain words are, based on their syntactic dependencies [9]. Other researchers working on WSD thought to use graph-like approaches such as PageRank [10,11] to map relationships between pages and words. This later developed further, as researchers started to correlate word meanings as vertices and the edges as the semantic relationships derived from WordNet to construct a disambiguation graph [12]. The disambiguation graph provides a context and understanding of words in relation to each other and can offer a deeper insight into the likelihood of determining which words fit which contexts. Similarly to PageRank, a different approach of considering the term frequency-inverse document frequency (TF-IDF) was utilized to relate context with frequency of occurrence [13,14]. TF-IDF is a common indexing method used for analyzing likelihood and has applications in areas such as web page retrieval and recommendation systems but can also be utilized to analyze text and the context of words. The authors Turtel and Shasha utilized TF-IDF for acronym disambiguation [15]. Similarly, the authors Li et al. utilized word embedding paired with TF-IDF to solve acronym disambiguation [16].

The third approach is to use a transformer model such as BERT (bidirectional encoder representations from transformers) [17,18]. Traditionally, recurrent neural networks using a long short-term memory have been deployed for natural language models. Recently BERT has replaced most of those networks. Essentially, BERT is an upgrade on recurrent neural network models, as it is able to take in information directionlessly. Previously, we demonstrated some of the power of BERT in its ability to generate abbreviation definitions [19] and further successfully tested it on ambiguous definitions [13]. The authors Daza et al. utilized a SloBERTa model with an additional single neural layer to tackle abbreviation disambiguation for Slovenian biographical lexicons [20]. The benefits of using BERT models include having access to differently trained variations on domain-specific data including, but not limited to, models such as RoBERTa [21], SciBERT [22], and AlBERT [23].

Within this scope, there are still many limitations that can be improved upon. The most salient problem is the lack of domain specific data and centralized data. In regards to proprietary and personal data, especially in the medical field, there is a lack of sufficient datasets that encompass all definitions and their multiple abbreviation forms. Additionally,

ambiguous definitions can exist across domains, with no accounting for these disparities. There has been some progress towards creating a consolidation of publicly available data, such as the database MeDal, which is composed of 14,393,619 medical articles and abstracts. The authors, Wen et al. utilized this dataset to pre-train several machine learning models. The results showed improvements towards being able to accurately define abbreviations [24]. Alternatively, the researchers Skreta et al. combined data sampling and reverse sampling (RS) to automatically create their dataset, without the need for human aid [25].

Abbreviations, with their convenience of use and their improvement of efficiency in all text-related fields, will continue to find prolific usage in professional contexts. Inevitably, this leads to their misunderstanding and misinterpretation, due to definition ambiguity and over-utilization. However, with insight into how abbreviations are commonly used and keeping to a standard of abbreviation guidelines, we could minimize these potential misreadings. In this paper, we introduce a novel way to generate ad hoc abbreviations, produce their definitions, and to reverse engineer their candidate definitions. Our work is still in the early stages, but we have found several interesting statistics, such as higher definition retrieval rates for abbreviations retaining 40% of their characters and trends such as a 93% omission rate for vowels. These preliminary results can be further utilized to lay foundations and create certain rules involving contraction-type abbreviations.

## 2. Materials and Methods

To fully tackle abbreviation disambiguation, the ideal scenario is to have a dictionary with all possible abbreviations and their definitions assigned. Realistically, this is not plausible, as new abbreviations and deviations are constantly being produced. However, it may be possible to generate all or close to all possible abbreviations that are likely to be used and assign them sufficient context to aid WSD and allow selecting the best possible candidate for expansion.

Currently, most of the work and data have focused on tackling acronyms. However, not much work has been done around contraction-based abbreviations. Contractions are a subset of abbreviations where, instead of a phrase being abbreviated, a single word is. Similarly to acronyms, contraction-type abbreviations can be related to sub-string matching, but with characters. Pennel et al. introduced a normalization model for contractions found in Twitter data [26]. Their method leans more towards a statistical model for detection and focuses on manual annotations to create reasonable abbreviations.

Traditionally, acronyms have unique characteristics, such as containing all or a majority of capital letters, annotated within parentheses, and having long-form definitions provided. Contractions tend to have traits that are more difficult for a computer to parse, such as having punctuation as part of the word, containing misspellings of words, and having a collection of letters that do not make for fluid pronunciation. Additionally, what makes this type of abbreviation harder to expand is the lack of a provided definition. The underlying goal of contraction expansion is to identify which long-form definition the substring could soundly be a part of.

Figure 1 shows our experimental design process. In the later sections, we will elaborate more on the data generation process, word selection, abbreviation process, and reconstruction of the definitions.

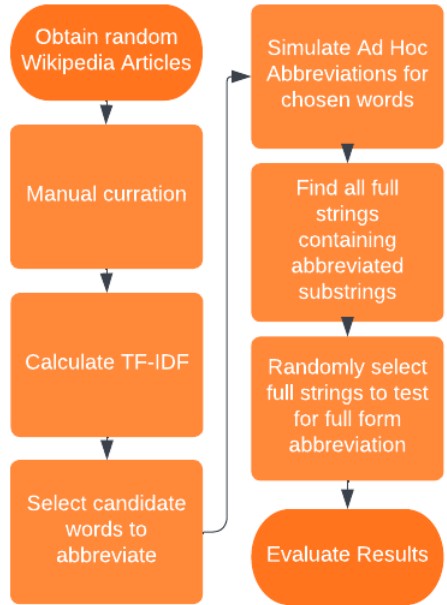

**Figure 1.** Experiment Flowchart.

## 2.1. Dataset

In order to simulate a larger amount of domains and introduce ambiguity, we wanted to create a corpus that introduces different topics. Initially, we intended to use the MeDal dataset, but the provided dataset contained a vast amount of errors, including typos and unidentifiable characters and symbols. Instead, we collected different text files, generated from Wikipedia articles. These were text documents of random Wikipedia pages, created by using the built in random article generator in the Wikipedia API for python. We simulated different domains by manually curating articles to have a diversity of domains and genres, with some small overlaps. Our final dictionary contained 35 different articles in various domains, including but not limited to autobiographies, education, music, technology (circuits), and film. There was no strict criteria for document acceptance, other than a soft attempt to maximize domain and genre diversity. We would like to argue that contraction-based abbreviations can be used with any part of speech (noun, verb, etc.), which is why we did not use a stricter method for accepting documents. In addition, document size did not play a role in our selection process, as we wanted to keep the simulated variance.

We then calculated the number of occurrences of each word and the document domain they occurred in. We kept everything in the corpus as it was presented, and introduced a few rules to select abbreviation candidates to be contracted. Let us denote the candidate abbreviation set as $C$.

For candidate words, we make a few modifications to our criteria:

1. `Frequency`—the amount of times the word that is intended to be abbreviated is used; it should be quite frequent to warrant the need to abbreviate. We decided on a minimum of seven occurrences;
2. `Length`—the length of the candidate word must be greater than five letters;
3. `Outliers`—unnatural words such as words with numbers were not included;
4. `Replacers`—abbreviations with replacement characters or symbols that were not found in the original long-form definition (ex: XFER with X in place of the word Trans) were omitted.

To further elaborate our justifications for each rule, we chose words with frequencies greater than seven occurrences, as this meant the term could potentially be used in more than one domain and yielded a higher contextual significance. On average, our occurrence rate was above 10 uses for words that could be found in at least 2 documents. We deliberately did not choose any stop words, as they are generally uninformative [27] and have

commonly understood or established contractions already. We also excluded words that had a length less than five letters, as they would not require abbreviation. Our average word length was approximately eight letters. Often, in medical domains or fields with high Latin and Greek usage, such as chemistry, compounds and scientific names are abbreviated using numbers and Greek letters. We chose to omit these types of words as they have more domain specific rules and guidelines that cannot be generalized easily. Lastly, for candidate definitions, we chose to omit any symbolic representations for similar reasons, as they have their own specific guidelines to follow.

### 2.2. Ad Hoc Abbreviations

Since abbreviations have no strict rules for generation, different variations of the same word can occur. We wanted to investigate what characteristics may impact the generation of ad hoc abbreviations and whether or not guidelines can be established to promote better abbreviation generation. For each item in *C*, we utilized an online user submitted dictionary, https://www.abbreviations.com/, (accessed: 24 April 2023) to find the most common abbreviations for each word. We chose the top-five unique abbreviations for each item and created a dictionary *D* for them. It is important to note that not every definition had more than five unique abbreviations to choose from. Table 1 shows a subset of a long-form word and its candidate ad hoc abbreviations.

**Table 1.** Long-form definition with candidate contractions.

| Word | Instances | Candidate 1 | Candidate 2 | Candidate 3 | Candidate 4 | Candidate 5 |
|------|-----------|-------------|-------------|-------------|-------------|-------------|
| background | 18 | bg | bkgd | bkgnd | bk | bgr |
| additional | 13 | addl | addtl | addnl | | |
| primary | 10 | pri | pry | pr | prim | |
| broadcast | 10 | brd | brdcst | bcst | | |

From our various contractions, we needed to assign candidate definitions to them. To simulate potential ambiguous definitions, we needed to perform a dictionary look up for all the words containing the contraction letters. We utilized an online platform https://www.dcode.fr/en (accessed: 24 April 2023). This allowed us to input any substring and search a dictionary to find all words containing the substring.

From the returned substrings, we removed strings that did not start with the same letter as the abbreviation and randomly selected five potential definitions from the remainder. Among the five selected definitions, one was always the ground truth definition. If a substring that was selected had the same root, such as a plural form or different tense, we removed it and randomly chose a different word, if available. Additionally, not all abbreviations contained more than five candidate definitions. We noticed a positive correlation between the increase in substring length and the decrease in potential candidates. Table 2 shows a sample ad hoc abbreviation found in Table 1 and its potential long-form expansions using a dictionary look up.

After building up a corpus of potential abbreviations and their candidate long forms, we tested for ambiguity utilizing a TF-IDF model and BERT. However, in practice, the BERT model yielded poor results, largely due to the lack of definitions retained in the model. In a previous paper [19], we explored the use of BERT on acronym-type abbreviations, and it presented a 94% acceptance rate. Using the same method, but querying for contractions, the BERT model scored 0% for acceptance.

**Table 2.** Ad Hoc abbreviations and their candidate definitions.

|  | Candidate Definition 1 | Candidate Definition 2 | Candidate Definition 3 | Candidate Definition 4 | Candidate Definition 5 | Candidate 5 |
|---|---|---|---|---|---|---|
| bg | bag | begar | being | background | big | bgr |
| bkgd | backgated | backgrind | background | backsighted | | |
| bkgnd | backgrind | background | backgammoned | | | |
| bk | blink | break | back | black | background | |
| bgr | begar | bagger | bigger | background | burger | |

There are two major flaws with using the BERT model. The first issue is that the document containing the ground truth must be given to the model. Outside of training, BERT will not retain new information. The second issue is the model itself does not comprehend contractions. For example, when prompted for acronyms such as *"What does RAM mean?"* the model will return *random access memory*. However, when prompted for contractions, an example prompt would include *"What does bg mean?"* and its corresponding result would often return junk or an entire snippet of the document such as *"TECHNICAL FIELD The present invention relates to a wireless local access network (WLAN), and more particularly, to a channel access mechanism for a very high throughput (VHT) WLAN system and a station supporting the channel access mechanism. BACKGROUND ART . . ."*.

For the remainder of our WSD for definition expansion, we will use TF-IDF. Since each document can be roughly treated as its own category or domain (after curation), the WSD assumption would correlate with the correct definition in its respective domain.

## 3. Results

Our dictionary lookup utilizing https://www.dcode.fr/en generated over 1000 candidates. Table 3 details the different candidate definitions chosen, as well as the number of potential candidate definitions to chose from, what the ratio of removed letters were in the contraction, and statistics on the vowel removal. We noticed a drop in the number of potential candidates to choose from, depending on the ratio of the length of the substring to the actual word. Naturally, the more characters that are retained, the less likely it can generate extra options that can be ambiguous. To summarize, we found that, on average, utilizing 50.769% of the characters to abbreviate reduced the candidate options from over 1000, to less than 100. The lowest occurrence we found was when utilizing 40% of a string reduced candidates to 53.

Out of the 58 vowels that were found in the long-form definitions, not including the starting letter, we found that 93.103% of vowels were removed by the ad hoc abbreviations. An interesting statistic to mention is that none of the abbreviations' low candidate selections kept any non-first-letter vowels. This may imply that vowels contain a low amount of information in contraction-type abbreviations. Conceptually, this makes sense, as there are usually only five potential vowels to chose from.

Testing for ambiguity between definitions, we calculated the TF-IDF for each definition per category. Candidate definitions that do not appear in our corpus scored 0. We have provided the link to our TF-IDF calculations, as well as the curated text files used to build our dictionary (https://github.com/choivsh/contraction-data). Among our abbreviations, we found six abbreviations that had ambiguous definitions. These six abbreviations included adj, ad, mods, pry, py, and prim. Examples of their ambiguous definitions included words such as modules, modification, primary, and proxy.

We were only interested in whether or not TF-IDF was strong enough to disambiguate the definitions, so we compared the document probabilities of the overlapped data. Essentially, any document that did not overlap would guarantee the ground truth definition was the only contender. For the overlapping documents, we found that only one out of seven categories selected the correct definition, but this was significant (the difference in probability was greater than 0.01). For the remaining six categories that were marked incorrect, the probability difference was less than 0.01, making the results lean towards being inconclusive. Table 4 is a subset of the probability calculations based on our TF-IDF model.

**Table 3.** Candidate definition statistics.

| | Definition 1 | Definition 2 | Definition 3 | Definition 4 | Definition 5 | # Candidate Words) | Letter Ratio | Letters Removed | Vowels Found | Vowls Kept | Vowels Removed |
|---|---|---|---|---|---|---|---|---|---|---|---|
| bg | bag | begar | being | background | big | 1000+ | 0.2 | 8 | 3 | 0 | 3 |
| bkgd | backgated | backgrind | background | backsighted | | 53 | 0.4 | 6 | 3 | 0 | 3 |
| bkgnd | backgrind | background | backgammoned | | | 17 | 0.6 | 4 | 3 | 0 | 3 |
| bk | blink | break | back | black | background | 1000+ | 0.2 | 8 | 3 | 0 | 3 |
| bgr | begar | bagger | bigger | background | burger | 1000+ | 0.3 | 7 | 3 | 0 | 3 |
| addl | addle | adducible | additional | addressible | adjudgeable | 860 | 0.4 | 6 | 4 | 0 | 4 |
| addtl | additional | adjudicational | adductively | autodidactically | addictionologists | 47 | 0.5 | 5 | 4 | 0 | 4 |
| addnl | addental | additional | antroduodenal | addictionologist | | 65 | 0.5 | 5 | 4 | 0 | 4 |
| adj | adjunct | adjutant | adjacent | adjust | adjectives | 582 | 0.5 | 3 | 1 | 0 | 1 |
| ad | advance | abide | added | admit | ahead | 1000+ | 0.333333333 | 2 | 3 | 0 | 3 |
| alloc | allocate | allowance | allothetic | allocrite | allogenic | 1000+ | 0.5 | 5 | 4 | 1 | 3 |
| mods | modish | modules | modification | modelss | micropods | 1000+ | 0.307692308 | 9 | 6 | 1 | 5 |
| pri | peril | pride | prima | primary | praise | 1000+ | 0.428571429 | 4 | 2 | 1 | 1 |
| pry | proxy | prayer | poetry | purify | primary | 1000+ | 0.428571429 | 4 | 2 | 0 | 2 |
| pr | prom | prop | proxy | primary | professor | 1000+ | 0.285714286 | 5 | 2 | 0 | 2 |
| prim | primal | pluralism | prelimits | primary | perimeter | 1000+ | 0.571428571 | 3 | 2 | 1 | 1 |
| brd | broadcast | birds | breed | broiled | bromide | 1000+ | 0.333333333 | 6 | 3 | 0 | 3 |
| brdcst | breadcrust | broadcast | | | | 46 | 0.666666667 | 3 | 3 | 0 | 3 |
| bcst | backseat | backstep | buckshot | biochemist | broadcast | 1000+ | 0.444444444 | 5 | 3 | 0 | 3 |

**Table 4.** Probability of selecting definition.

| Target Definition | Candidate Definition | Category Number | Target Probability | Candidate Probability |
|---|---|---|---|---|
| adjust | adjacent | 2 | 0.00854 | 0.02135 |
| adjust | adjacent | 14 | 0.00311 | 0.00623 |
| adjust | adjacent | 16 | 0.00146 | 0.00439 |
| advance | added | 13 | 0.02537 | 0.00260 |
| modification | modules | 14 | 0.00357 | 0.00936 |

## 4. Discussion

Overall, this study can be generalized as a proof of concept of creating a dictionary containing various ad hoc abbreviations. The results in our sample demonstrated some causal correlations between which letters can be truncated while retaining enough information to narrow the definition search further. If coupled with categorical tagging, machines are able to disseminate and replace contraction-type abbreviations with their long-form definitions. We would like to discuss some improvements, as well as future work that this line of research could be extended to.

First, we would like to explore some limitations of our model. The candidate definitions that came from the dictionary lookup yielded over 1000 visible entries and permutations. As there was no available API for https://www.dcode.fr/en, we did not have an effective way of reducing the candidate definitions by removing any definition that did not start with the same starting character (e.g., bg could return: airbag). This may artificially inflate the number of candidates returned; however, the overall trend of increasing substring characters would still reduce the amount of candidates returned. A potential solution would be to store a standardized English dictionary and query with the criteria needed to decipher.

Second, we did not have an effective way of automatically generating ad hoc contractions. While it is possible to permute different iterations of the candidate string, we argue that having real-life data and utilizing user submitted abbreviations more accurately simulates an ad hoc nature. Utilizing this method, we could lay a foundation and extract a few important rules and assumptions based on the findings, to implement a model in the future that reduces the amount of nuisance and noisy candidate abbreviations. However, an argument could be made to generate those permutations, to see whether or not a pattern

could be discovered that performs better than the naturally accepted abbreviations made by humans.

Following this train of thought, we could build a rule-based model to anticipate and predict the probability of the next letter in the substring and reverse engineer the best candidate contraction for each word. This could be closely related to a HMM used for acronyms. In future work, we plan to use further pattern recognition techniques to extract more characteristics that make up a contraction and determine a stricter set of rules, to reduce the amount of ambiguity in definition finding. Currently, we have established that maintaining 40% of the letters in a contraction-type abbreviation may be a good baseline to filter candidate definitions. We are considering investigating a word's features, such as the context and part of speech, as this may give indications that can eliminate potential candidates. This would allow for less strict guidelines, in the hope that it is possible to reduce the ratio of letters needed.

One additional note is the relative size of our dataset. We have demonstrated as a proof of concept that we can build a small dictionary of terms, and this model could be scaled further, to create a larger corpus for better results. Context is usually given to abbreviations, making their domain typically known. Specialized dictionaries could be made to help target more specific domain abbreviations. When scaled for generalized abbreviations, a common practice in analyzing TF-IDF values is to take the negative log of the percentages to yield a more easily observed significance.

## 5. Conclusions

Based on our findings, we were able to artificially simulate some ad hoc abbreviations that could potentially be used in everyday language. From this, we were able to discover some interesting statistics about contraction-type abbreviations, such as vowel usage and removal, and the impact of string length ratio. This area could be further extended to discovering patterns within substrings that can minimize the number of potential candidate definitions. Additionally, we ran tests on these ad hoc abbreviations and generated candidate definitions, which proved to be difficult for the TF-IDF model to evaluate. Our proof of concept shows that contraction-typed abbreviations behave differently than acronyms, and minimizing the potential candidate definitions would be a step in reducing their ambiguity. In future work, we plan to automate the current steps, to scale the concept and uncover more potential patterns for generating easily understood and retrievable contraction-type abbreviations. In addition, we plan to localize this concept and apply it in specific domains, such as the biomedical field, to fully test our findings when the context is much narrower.

Table A1 is a shortened excerpt from the main TF-IDF calculations (the calculations are available on GitHub link provided in later section). Each document category has its own summary of probability (some categories are named, some are numerical, depending on how the API's metadata named the category). We tallied the total number of occurrences as well. In total we had 35 categories to choose from, with 612 unique words detected in the dictionary.

**Author Contributions:** Conceptualization, S.C. and K.T.; methodology, S.C. and K.T.; software, S.C.; validation, S.C. and K.T.; formal analysis, S.C.; investigation, S.C. and K.T.; resources, S.C.; data curation, S.C.; writing—original draft preparation, S.C.; writing– review and editing, S.C. and K.T.; visualization, S.C. and K.T.; supervision, K.T. All authors have read and agreed to the published version of the manuscript.

**Funding:** This research received no external funding.

**Data Availability Statement:** Dataset used can be found here at: https://github.com/choivsh/contraction-data.

**Acknowledgments:** We would like to acknowledge the open resources provided by Wikipedia, https://www.dcode.fr/en, and https://www.abbreviations.com/, which made it possible to generate the data for our experiments.

**Conflicts of Interest:** The authors declare no conflict of interest.

## Abbreviations

The following abbreviations are used in this manuscript:

| | |
|---|---|
| BERT | Bidirectional Encoder Representations from Transformers |
| WSD | Word Sense Disambiguation |
| TF-IDF | Term Frequency-Inverse Document Frequency |
| HMM | Hidden Markov Models |
| POS | Parts of Speech |
| RS | Reverse Sampling |
| API | Application Program Interface |

## Appendix A

**Table A1.** TF-IDF Sample Data.

| Document ID | Dictionary Words | | | | | | | | |
|---|---|---|---|---|---|---|---|---|---|
| | appearances | appears | appended | appletalk | applets | appliances | applicability | applicable | application |
| 10 | 0 | 0 | 0.005293352 | 0 | 0 | 0 | 0 | 0.005059638 | 0 |
| 12 | 0 | 0 | 0.009081262 | 0 | 0 | 0 | 0 | 0.004340151 | 0 |
| 13 | 0 | 0.005329779 | 0.003649573 | 0.005949893 | 0.005949893 | 0.011899787 | 0 | 0 | 0.031551542 |
| 14 | 0 | 0 | 0.002396322 | 0 | 0 | 0 | 0 | 0 | 0.007533401 |
| 15 | 0 | 0 | 0 | 0 | 0 | 0 | 0 | 0.010297847 | 0.005080364 |
| 16 | 0 | 0 | 0 | 0 | 0 | 0 | 0 | 0 | 0.002266803 |
| 17 | 0 | 0 | 0.006081679 | 0 | 0 | 0 | 0 | 0.003875439 | 0.00637306 |
| 18 | 0 | 0 | 0.00577253 | 0 | 0 | 0 | 0.002107528 | 0.004138245 | 0.002268413 |
| 2011_Saransk_Cup | 0 | 0 | 0 | 0 | 0 | 0 | 0 | 0 | 0 |
| 23 | 0 | 0 | 0 | 0 | 0 | 0 | 0 | 0 | 0 |
| 4 | 0.001940913 | 0 | 0.002381053 | 0 | 0 | 0 | 0 | 0 | 0.004678373 |
| 420 | 0 | 0 | 0 | 0 | 0 | 0 | 0 | 0 | 0 |
| 519 | 0 | 0 | 0 | 0 | 0 | 0 | 0 | 0 | 0.096982195 |
| 6 | 0 | 0 | 0 | 0 | 0 | 0 | 0 | 0.023719197 | 0.00097514 |
| 699 | 0 | 0 | 0 | 0 | 0 | 0 | 0 | 0.002544519 | 0.018829786 |
| 7 | 0 | 0 | 0 | 0 | 0 | 0 | 0.002803288 | 0 | 0.006034572 |
| 795 | 0 | 0 | 0.001251368 | 0 | 0 | 0 | 0 | 0.002392234 | 0.000983492 |
| 8 | 0 | 0 | 0 | 0 | 0 | 0 | 0 | 0.006762265 | 0.001010943 |
| 9 | 0 | 0 | 0 | 0 | 0 | 0 | 0 | 0 | 0.001361715 |
| … | … | … | … | … | … | … | … | … | … |
| Occurrances | 1 | 2 | 8 | 1 | 1 | 1 | 2 | 9 | 14 |

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
