# Peer review of "Findings on Ad Hoc Contractions"

_information, doi:10.3390/info14070391_

Round 1

Reviewer 1 Report

My comments:

Point 1: It is suggested that the actual contribution and significance of this study should be emphasized. The practical effect of this study is to recommend the generation of abbreviation results, or statistical characteristics of the law of abbreviation. The author can elaborate further.

Point 2:  In Section 2, A technical roadmap of methodology can be added. At present, the text is more abstract.

Point 3: In Section 3, using a dictionary query to obtain abbreviated results, what is the evaluation method for experimental results. Is it necessary to evaluate the accuracy of abbreviation results. Also, because there are various candidates for abbreviation results, whether the candidates need to be sorted again.

Author Response

Thank you for your review. We have modified the following items to address the issues mentioned.

  • Made major revisions to word usage to more accurately reflect certain aspects of the work
  • Included a flowchart in the methodology to help guide readers.
  • The evaluation is based on the found probability of a definition based on querying the TF-IDFs that were calculated. We changed the wording in section 3 to better reflect that.

Reviewer 2 Report

Summary

This paper highlights the prevalence of abbreviations in various aspects of communication, such as business names, handwritten notes, online messaging, and professional domains. It points out that the abundance of abbreviations, along with the continuous introduction of new ones, leads to overlapping and ambiguous situations, resulting in a loss of clarity in documents. The authors state that they have conducted a reverse engineering process to understand how these ad hoc abbreviations are created. They also present some preliminary statistics on factors that make abbreviations easier or harder to define. Furthermore, the authors mention generating potential definitions for these abbreviations, which posed challenges for a word sense disambiguation model to accurately select the correct definition.

Indeed, acronym usage is increased nowadays and methods for automatically providing their definitions are very useful, so this work is highly relevant. The title of the work and the abstract do not accurately describe the work done and what is described in the results section. These need to be accurately reported in the text.

One part of this work that needs some work is the experimental description which should provide more insights on the dataset artificially created, the method used to create the dictionary from the files provided at https://github.com/choivsh/contraction-data/tree/main/datagen_output. Moreover, the authors state at line 129 that “Instead, we collected 35 different text files, generated from Wikipedia articles.”, how did you select these 35 text files randomly when you have a set of strict rules applied? Manually, automatically? Curated? What are some quantitative characteristics of these wikipedia documents? Size, number of named entities, verbs/nouns? Did you perform an analysis on these documents? Connecting this with line 127, how do you make sure that these articles are from different domains?

Next, it is very difficult to follow the dataset creation process. It would be very beneficial to provide a graphical way to represent this or with a pseudo-algorithm. In the keywords and the introduction the authors mention that they will also use BERT for testing their premise, but in line 192 they state that “However, in practice, the BERT model yielded poor results largely due to the lack of definitions retained in the model.” and present no results on their findings, which could be valuable for the readers.

The authors should extend their literature review section, since there are a number of interesting and similar efforts for acronym and abbreviation disambiguation, and it would be important for the interested reader to know the difference with other works. For example the following papers are on the same topic and follow a similar IR technique:

1) Chao Li, Lei Ji, Jun Yan(2015), Acronym Disambiguation Using Word Embedding

2) Benjamin D. Turtel, Dennis Shasha. Acronym Disambiguation

3) Angel Daza, Antske Fokkens, Tomaž Erjavec (2022). Dealing with Abbreviations in the Slovenian Biographical Lexicon

This work could also benefit from similar IR methods in conjunction with word knowledge (commonsense) used in Geographic focus identification and disambiguation see for example relevant work from C. Rodosthenous and L. Michael (2019). Using Generic Ontologies to Infer the Geographic Focus of Text and also check the literature review there.

In terms of writing and presentation, the authors need to revisit their work and correct grammatical and syntactical mistakes. Please consider moving Table 1 in the Appendix as it does not offer much in the understanding of the method where it is placed.

Some other comments:

Please explain why “Because there was no API, we did not have an effective way of reducing the candidate definitions by removing any definition that did not start with the same starting character (eg: bg could return : airbag)”. No API for what?

At line 181, the authors state“ We utilized an online platform decode.fr.”. This site is not available and it is not clear what type of data and methods it offers. Maybe the authors refer to https://www.dcode.fr ? Please provide more information on the tool you used if that is the case.

At line 216, the authors state “Amongst our abbreviations we found 6 abbreviations that had ambiguous definitions. ”. Since these are only 6 you could explicitly mention them and use them as example of this ambiguity.

Another concern raised for this research is the small dataset size. The authors mention at lines 261-265 that “We have demonstrated as a proof of concept we can build a small dictionary of terms and this model can be extended further to create a larger corpus for better results. With a larger corpus however, the probabilities will be stretched out thinner and would require an additional conversion metric to evaluate the results.” which contradicts with the fact that this work can be generalized.

To conclude, this work is in preliminary stages and further experimentation is in order to provide results towards the claim provided in the abstract and introduction. Furthermore, major revisions are needed to make this work publishable and of interest to the relevant community.

Included in the general comments section.

Author Response

Thank you for the review. We have made the following changes addressing the points.

  • We included further elaboration on the dataset generation process and additional details in section 2.1.

    In short, the documents were manually curated to maximize genre/domain diversity with no stricter criteria for acceptance. We included some general domain examples.

  • The use of BERT was trialed and it did not present adequate results. We expanded our explanation as to why this part was omitted by providing an example and comparing it with previous results when used on acronyms. In addition we replaced BERT from the keywords as it is not the major focus of this research.  

  • Additional literature review was included.

    The table was previously requested by a reviewer and we have now moved it to the appendix.

  • Further clarity on the API was added.

  • The website link was corrected to https://www.dcode.fr/en.

  • The 6 abbreviations are now listed with a few examples of their ambiguous definitions.

  • We rephrased the sentence to explain how to scale the model. Adding more data and domains will naturally create small percentages for the TF-IDF values. We changed the wording to reflect the method in converting small percentages by taking its logarithm to more show easily show significance.

  • Grammatical and syntax changes were made throughout the paper with the assistance of extra reviewers for English.

Round 2

Reviewer 2 Report

The authors have revised their work and it is now more readable. Necessary clarifications were provided in the text. This work is in a preliminary stage and more experiments are needed to come to safe conclusions, as the authors also state. This does not mean that readers are not interested in this work, but it should be made clear that it is indeed in its early stages. The changes made in the title reflect this, but the authors should also make that more clear in the introduction.

There are still several typos and syntactic mistakes which must be corrected before publication:

Line 37: Of what an acronyms and abbreviations are

Line 62: and selects the one [which] makes the most sense.

Line 131: to tackle initialism- more com-

Line 151-152: we wanted [to] create a corpus that introduces different topics.

Line 241: of potential candiate definitions to chose from

Line 253: Five potential vowels to chose from

Paragraphs 91-101 and 102-112 have the same content with just minor differences. Please check and revise.

Author Response

Hello,

Thank you for your follow up review. We have made the following changes:

  • Included an additional statement in the introduction to emphasize the early stages of the work
  • Revised the English in the lines mentioned. Line 131 simplified further.